Developmental toxicity from exposure to various forms of mercury compounds in medaka fish (Oryzias latipes) embryos

Dong Wu wu.dong@duke.edu 1 2
Liu Jie 3
Wei Lixin 4
Jingfeng Yang 1
Chernick Melissa 2
Hinton David E. 2
1 Inner Mongolia Provincial Key Laboratory for Toxicants and Animal Disease, College of Animal Science and Technology, Inner Mongolia University for the Nationalities , Tongliao , China
2 Nicholas School of the Environment, Duke University , Durham , NC , United States
3 Zunyi Medical College, Department of Pharmacology , Zunyi , China
4 Department of Tibetan Medicine, Northwest Institute of Plateau Biology, Chinese Academy of Sciences , Xining , China
Stegeman John
Electronic publication date: 2016 Aug 23
Publication date: 2016
Volume: 4
Electronic Location ID: e2282
Received 2016 Jan 15; Accepted 2016 Jul 2
Copyright: ©2016 Dong et al.
Copyright year: 2016
Copyright holder: Dong et al.
License: This is an open access article distributed under the terms of the Creative Commons Attribution License, which permits unrestricted use, distribution, reproduction and adaptation in any medium and for any purpose provided that it is properly attributed. For attribution, the original author(s), title, publication source (PeerJ) and either DOI or URL of the article must be cited.
License URL: https://creativecommons.org/licenses/by/4.0/

Keywords: MeHg; HgCl2; α-HgS (Zhu Sha, cinnabar); β-HgS (Zuotai); Medaka; Developmental toxicity; Heme oxygenase-1; Mercury; Metallothionein

Funding: The National Natural Science Foundation of China 21267015 81360508 21567019 Natural Science Foundation of Inner Mongolia Autonomous Region of China 2015MS0804 Project of Scientific and Technical Innovation of Inner Mongolia Autonomous Region Research Program of Science and Technology at Universities of Inner Mongolia Autonomous Region NJZY11202 This study was supported by the National Natural Science Foundation of China (21267015, 81360508, 21567019); Natural Science Foundation of Inner Mongolia Autonomous Region of China (2015MS0804), the Project of Scientific and Technical Innovation of Inner Mongolia Autonomous Region (2015–2016) and Research Program of Science and Technology at the Universities of Inner Mongolia Autonomous Region (NJZY11202). The funders had no role in study design, data collection and analysis, decision to publish, or preparation of the manuscript.

==============================
This study examined developmental toxicity of different mercury compounds, including some used in traditional medicines. Medaka (Oryzias latipes) embryos were exposed to 0.001–10 µM concentrations of MeHg, HgCl2, α-HgS (Zhu Sha), and β-HgS (Zuotai) from stage 10 (6–7 hpf) to 10 days post fertilization (dpf). Of the forms of mercury in this study, the organic form (MeHg) proved the most toxic followed by inorganic mercury (HgCl2), both producing embryo developmental toxicity. Altered phenotypes included pericardial edema with elongated or tube heart, reduction of eye pigmentation, and failure of swim bladder inflation. Both α-HgS and β-HgS were less toxic than MeHg and HgCl2. Total RNA was extracted from survivors three days after exposure to MeHg (0.1 µM), HgCl2 (1 µM), α-HgS (10 µM), or β-HgS (10 µM) to examine toxicity-related gene expression. MeHg and HgCl2 markedly induced metallothionein (MT) and heme oxygenase-1 (Ho-1), while α-HgS and β-HgS failed to induce either gene. Chemical forms of mercury compounds proved to be a major determinant in their developmental toxicity.

Introduction

Mercury-based traditional medicines are an important consideration in public health of specific countries. For centuries, mercury has been used as an ingredient in diuretics, antiseptics, skin ointments and laxatives, and more recently, as a dental amalgam and as a preservative in some vaccines (Clarkson, Magos & Myers, 2003; Liu et al., 2008). In traditional Indian Ayurvedic (Kamath et al., 2012), Chinese (Pharmacopeia of China, 2015) and Tibetan medicines (Chen et al., 2012; Kan, 2013; Li et al., 2014; Wu et al., 2016), mercuric sulfides are frequently included in the treatment of various disorders, with the result that health concerns for public safety are increasing (Liu et al., 2008; Kamath et al., 2012). This form of mercury, from the naturally occurring minerals, cinnabar and metacinnabar, typically undergoes purification and preparation prior to use (Kamath et al., 2012; Li et al., 2016). Zuotai is primarily composed of β-HgS (metacinnabar) while cinnabar (Zhu Sha) is α-HgS (Li et al., 2016; Wu et al., 2016). Only mercury sulfides are used in traditional remedies because they are considered to be safe at clinical dose levels (Liu et al., 2008). The Chinese Ministry of Health has closely monitored mercury contents in these medicines and publishes allowable doses (0.1–0.5 g/day) in the Pharmacopeia of China (Liang & Shang, 2005; Liu et al., 2008). However, these doses can be considerably higher than what is considered to be safe in Western countries (Liu et al., 2008) (see Table S1). Inorganic mercury chloride (HgCl2) and organic methylmercury (MeHg) forms are highly toxic and never used in these treatments (Kamath et al., 2012). This distinction is important because it is the total mercury content rather than specific chemical forms that are commonly used to assess risk of traditional medicines, and this approach may be inaccurate.

Mercury is categorized as a nonessential metal with no biological function but concentration-dependent toxicity (Sfakianakis et al., 2015). Envrionmental transformation renders mercury of increased toxicologic relevance. First, the release of mercury vapor (Hg0) occurs following evaporation from water, soil, volcanic eruption/ash, and following certain industrial practices such as pulp and paper production, metal mining, and coal, wood and peat burning (Morel, Kraepiel & Amyot, 1998). Inorganic mercury is converted to MeHg by anaerobic bacteria present in sediments of fresh and ocean water (Liu, Goyer & Waalkes, 2008). This step is key for methylation and eventual bioaccumulation (Morel, Kraepiel & Amyot, 1998), affecting reactivity of mercury species as well as their concentration, lipid-permeability, and assimilation efficiency (Morel, Kraepiel & Amyot, 1998; Klaassen, 2001). Biomagnfication of mercury occurs with consecutive passage up the food chain (Morel, Kraepiel & Amyot, 1998; Authman et al., 2015). Because of their trophic positions as apex- or mesopredators, certain fish may contain high levels of mercury (Craig, 2003), and their consumption is the major route of human exposure to MeHg (Karimi, Fitzgerald & Fisher, 2012; Sheehan et al., 2014) (Table S1). In addition to the above dietary exposure, inorganic mercury exposure can occur via inhalation of mercury vapor from the chlor-alkali industry, heat extraction of gold from amalgam, and industrial discharge as Hg2+ (Liu, Goyer & Waalkes, 2008). While this awareness has led to the use of various fish species to investigate toxicity of mercury, less attention has been given to developmental toxicity.

Fish tissues have a high bioaccumulation capacity and are sensitive indicators of mercury pollution. Ingested mercurials are bound, stored, and redistributed by the liver and can be retained for long periods (Raldúa et al., 2007; Authman et al., 2015). More recently, in laboratory model fish species, early life exposures to inorganic mercury and MeHg have resulted in deformities, with eye, tail, and finfold alterations (Samson & Shenker, 2000). Advantages of these early life stage models are their low cost, rapid assessment, higher throughput, and easy determination of abnormalities. Medaka (Oryzias spp.) have been shown to be relatively sensitive to heavy metal exposure, including mercury (Dial, 1978; Ismail & Yusof, 2011; Mu et al., 2011). Their wide salinity tolerance and the development of marine models have led to a variety of studies of metal toxicity (Inoue & Takei, 2002; Chen et al., 2009; Mu et al., 2011). However, these studies have not tested traditional medicines (i.e., permutations of mercury ore, cinnabar).

Japanese medaka (Oryzias latipes) is a freshwater aquarium model fish with transparent embryos that allow for evaluation in ovo (Iwamatsu, 2004) as well as having a variety of molecular tools available (Cheng et al., 2012). Measuring gene expression is useful in identifying treatment-induced changes and mechanisms of action following exposure (Fielden & Zacharewski, 2001). Heme oxygenase-1 (Ho-1) is sensitive to a wide range of toxicants and has a protective role in the case of oxidative stress (Voelker et al., 2008; Weil et al., 2009). Metallothioneins (MT) are cysteine-rich, metal binding proteins that detoxify excess heavy metal ions and play a general role in antioxidant defense (Woo et al., 2006). Their expression has been shown to increase in a concentration-dependent manner in the presence of heavy metal contaminants; as such, they are considered to be a good biomarker for metal exposure in aquatic invertebrates (Amiard et al., 2006), laboratory model fish (Woo et al., 2006), and free-ranging populations of fish (Chan, 1995). These results follow closely with findings by Wu et al. (2016) in male Kunming mice exposed to organic-, inorganic mercury, and traditional medicines. The expression of these genes provides a way for us to evaluate mercury toxicity in medaka with the possibility of identifying mechanisms and commonalities with higher animal models. In this study, we determined the feasibility of using the medaka embryo assay as a tool to detect and compare developmental toxicity potentials of various forms of mercury (α-HgS, β-HgS, HgCl2, and MeHg); we assessed them for mortality, morphological changes, and toxicity-related gene expression.

Materials and Methods

Mercury compounds

HgCl2, MeHg (in the form of CH3HgCl), and α-HgS were obtained from Sigma-Aldrich (St. Louis, MO). Zuotai (hereafter referred to as β-HgS) was provided by the Northwest Institute of Plateau Biology, Chinese Academy of Sciences (Xining, China).

Medaka culture and embryo collection

Orange-red (OR) medaka (Oryzias latipes) were maintained at Duke University, Durham, NC, USA in an AHAB system (Pentair Aquatic Eco-Systems, Apopka, FL, USA) under standard recirculating water conditions. Brood stocks were housed in a charcoal-filtered, UV-treated water at 24 ± 2 °C with pH 7.4 and a light:dark cycle of 14:10 h. Dry food (Otohime β1; Pentair Aquatic Eco-Systems) was fed three times per day with supplementation of Artemia nauplii (90% GSL strain, Pentair Aquatic Eco-Systems) during the first two feedings. Embryos were collected by siphoning approximately 30 min after feeding, cleaned by rolling on a moistened paper towel, examined under a dissecting microscope (Nikon SMZ1500, Nikon Instruments, Inc., Melville, NY), and stage 10 embryos (6–7 h post fertilization (hpf)) were selected for experiments (Iwamatsu, 2004; Kinoshita et al., 2009) to represent an early exposure window (Villalobos et al., 2000; González-Doncel et al., 2008). Breeding colony maintenance, embryo collection, and experimental design followed animal care and maintenance protocols approved by the Duke University Institutional Animal Care and Use Committee (A062-15-02 and A031-15-01).

Experimental design

The experiment was conducted over a 10-day interval, from early blastula (stage 10) to hatching (Iwamatsu, 2004). All mercury compounds were dissolved in DMSO and in addition α-HgS and β-HgS were sonicated as described in Liu et al. (2008) and He, Traina & Weavers (2007) to increase solubility. Stocks were added to the wells of 6-well tissue culture plates (Corning, VWR International) at 1:1,000 dilutions in 5 mL 0.1% (w/v) artificial seawater (ASW) to obtain final concentrations of 0 (Control), 0.001, 0.01, 0.1, 1, and 10 µM, with a final DMSO concentration in each ≤0.1%. A total of 10–18 embryos were placed in each well, with three wells per mercury compound concentration, and solutions were not renewed during the course of the experiment. DMSO controls were chosen based on previous studies showing it does not contribute to toxicity (e.g., Dong, Matsumura & Kullman, 2010; Dong et al., 2014). Concentrations were chosen based on preliminary range finding assays in the same design that produced developmental abnormalities without leading to 100% mortality. Plates were incubated at 25 ± 2 °C on a 14:10 h light:dark cycle.

Mortality, hatching, growth, and teratogenesis

Embryos were observed daily under a dissecting microscope for mortality, hatching, delayed growth, and teratogenic effects. The latter included skeletal malformations, pericardial edema, decreased pigmentation of eyes, and swim bladder inflation or lack thereof. The Iwamatsu (2004) atlas was used to identify timing of events in organogenesis and hatching. Mortality was defined as any embryo with a brown, opaque chorion or any embryo with a non-beating heart. Hatching was defined as complete emergence from the chorion. Embryos that did not hatch by 10 days post fertilization (dpf) were considered dead. The experiment was terminated at 10 dpf and fish were euthanized by an overdose of MS-222 (Dong, Matsumura & Kullman, 2010; Colton et al., 2014).

RNA extraction and real-time PCR

A separate, identical exposure was run using the following concentrations: control, MeHg (0.1 µM), HgCl2 (1 µM), α-HgS (10 µM), and β-HgS (10 µM). These were the highest concentrations of each compound that yielded sufficient embryo numbers for RT-PCR analysis (n = 3, 15 embryos pooled per sample). Embryos were collected at 3 days post exposure, a time point selected based on survivorship in each treatment (Fig. 1). Embryos were homogenized with 1 ml of RNAzol using a stainless steel Polytron homogenizer (Kinematica, Newark, NJ). Following homogenization, total RNA was isolated as described in Dong, Matsumura & Kullman (2010). RNA quantity was determined using a NanoDrop ND-1000 spectrophotometer (ThermoScientific) and 260/280 ratios. Total RNA (500 ng) was reverse transcribed using High Capacity cDNA Reverse Transcription Kit (Applied Biosystems, Grand Island, NY). The following medaka specific RT-PCR primers were designed using Primer3 software and synthesized by Integrated DNA Technologies (Skokie, IL): Metallothionein (MT, AY466516, forward primer 5′-CTGCAAGAAAAGCTGCTGTG-3′, reverse primer 5′-GGTGGAAGTGCAGCAGATTC-3′); heme oxygenase-1 (Ho-1, AB163431, forward primer 5′-TGCACGGCCGAAACAATTTA-3′, reverse primer 5′-AAAGTGCTGCAGTGTCACAG-3′), and β-actin (S74868, forward primer 5′-GAGTCCTGCGGTATCCATGA-3′, reverse primer 5′-GTACCTCCAGACAGCACAGT-3′). The cDNA was amplified with SYBR Green PCR Master Mix (Applied Biosystems, Grand Island, NY). RT-PCR reaction conditions were 95 °C for 15 min followed by 40 cycles of 95 °C for 15 s and 60 °C for 60 s on the Applied Biosystems 7900HT instrument using their Sequence Detection System 2.0 software. For each sample, the threshold cycle (Ct) was normalized with β-actin of the same sample according to Chen et al. (2004). The amplification was calculated using the 2−ΔΔCT method (Livak & Schmittgen, 2001; Dong, Matsumura & Kullman, 2010).

Figure 1 Survival (%) of medaka embryos following exposure to MeHg (A), HgCl2 (B), α-HgS (C) and β-HgS (D) at 0 (control), 0.001, 0.01, 0.1, 1, or 10 µM (from stage 10 to 10 dpf).

Mortality was recorded daily as the percentage of nonviable individuals. Assays were run in triplicate with each data point representing the mean of n = 3 replicates of 10–18 embryos per replicate (±SD).

Statistical analysis

For each dpf, mean survival was calculated for each well and then used to calculate overall mean survival for each treatment group ±SD. Survival data were arcsine-square root transformed for ANOVA. RT-PCR data were normalized to β-actin expression and presented as mean ± SD. β-actin data were analyzed by a Grubbs Outlier test; outliers did not alter the results in subsequent tests and so were left in the analysis. For all measurements, one-way ANOVA followed by Tukey’s post-hoc test was used to assess the statistical significance among groups. A p ≤ 0.05 was considered to be statistically significant. All the data were analyzed using the SPSS 7.5 (SPSS Inc., Chicago, IL, USA).

Results

Mortality of medaka embryos after exposure to Hg compounds

Embryos exposed to 1 or 10 µM MeHg did not survive to 3 dpf (Fig. 1A). Whereas the mortality in these treatments was not statistically different from each other, they proved significantly higher than all other concentrations (p < 0.0001). The 0.1 µM group had significantly higher mortality than controls by 10 dpf (p < 0.05), increasing to 16.7% and to 26.7% mortality by 7- and 10 dpf, respectively (Fig. 1A). Those embryos exposed to 0.1 or 0.01 µM concentrations had increased mortality versus the 0.001 µM group (p < 0.05). While control mortality exceeded that of the 0.001 µM group, this was due to the loss of a single control individual (Table S2). HgCl2, while toxic, proved less so than MeHg. At 10 µM, all embryos died before 4 dpf. By 10 dpf, all other concentrations had 60% or higher survival, with all but the 10 µM statistically the same as the control (Fig. 1B). In comparison with the above, α-HgS and β-HgS were far less toxic and resembled survival levels seen in controls. For example, greater than 93% of embryos survived in all treatment groups by 10 dpf (Figs. 1C–1D).

Developmental toxicity

At 5 dpf, embryos exposed to either 0.1 µM MeHg or 1 µM HgCl2 showed malformations (Fig. 2). Delayed or arrested growth was also observed in 5 dpf embryos with MeHg and HgCl2 (Figs. 2B–2C). For example, 100% of individuals had reduced eye pigmentation, likely retina but further study is needed to confirm the site(s). By 10 dpf, 100% of hatched MeHg and HgCl2 exposed individuals showed uninflated swim bladders (Figs. 2E–2F) and associated swimming alterations (i.e., loss of buoyancy and equilibrium), but not α-HgS or β-HgS (Figs. 2G–2H). At 10 dpf, pericardial edema was observed in 80% of individuals in the 0.1 µM MeHg treatment but was absent in lower concentrations. This phenotype was also observed in 45% of the 1 µM HgCl2 exposed fish but was absent at lower concentrations. In severe cases, pericardial edema resulted in a tube heart in which expected anatomical positioning of heart chambers was absent (Figs. 2B–2C). No such edema was observed in α-HgS and β-HgS treatments. At 10 dpf, we observed bent body axis, surficial edema involving the skin above the inner ear, and a single cell mass projecting from the dorsal skin (Fig. 2F). In general, MeHg was observed to cause more severe and higher rates of deformity. No developmental abnormalities occurred with exposure to α-HgS or β-HgS by 5 dpf (not shown in figures) or by 10 dpf (Figs. 2G–2H).

Figure 2 Control morphology and common phenotypic alterations in embryos and larvae following exposure to mercury compounds: embryos at 5 dpf: (A) control; (B) 0.1 µM MeHg; and (C) 1 µM HgCl2.

Larvae at 10 dpf: (D) control; (E) 0.1 µM MeHg; (F) 1 µM HgCl2; (G) 10 µM α-HgS; and (H) 10 µM β-HgS. Arrows point to heart (h); e, eye; S, swim bladder. All images are at the same magnification, scale bar is 500 µm.

Metal toxicity-related gene expression

MeHg and HgCl2 increased MT mRNA expression by 4-fold and 5-fold over controls, respectively, while α-HgS (1.4-fold) and β-HgS (1.30-fold) had no appreciable effects (Fig. 3A). MeHg and HgCl2 increased Ho-1 mRNA expression by 6-fold and 2.3-fold over controls, respectively. α-HgS significantly increased Ho-1 expression (1.8-fold) over controls, but not to the degree of MeHg and HgCl2. β-HgS had no significant effects (1.4-fold) (Fig. 3B).

Figure 3 Analysis of gene expression in medaka embryos exposed to mercury compounds.

Medaka embryos were sampled from control, 0.1 µM MeHg, 1 µM HgCl2, 10 µM α-HgS, and 10 µM β-HgS treatment groups (n = 3, 15 embryos pooled per replicate) at 3 days post-exposure. Total RNA was extracted and subjected to RT-PCR analysis for metallothionein (MT) and heme oxygenase-1 (Ho-1) gene expression using β-actin expression as the reference. Data are mean ± SD. *significantly different from controls with p < 0.05.

Discussion

Mercury-based herbo-metallic preparations have been used in traditional medicines for thousands of years (Kamath et al., 2012) and continue to see usage today. Currently, the Pharmacopeia of China has 26 recipes that contain cinnabar (α-HgS). In Indian Ayurvedic medicine, Rasasindura, which is primarily composed of mercuric sulfides (α-HgS or β-HgS), is included in over 20 recipes (Kamath et al., 2012). In Tibetan medicine, Zuotai (β-HgS) is included in a dozen popular remedies (Kan, 2013; Li et al., 2014). By comparing MeHg and HgCl2 to α-HgS or β-HgS, we were able to assess compound related embryo toxicity and determine whether traditional medicines are toxicologically similar.

Numerous aquatic organisms have been studied with respect to the toxicity of mercury; however, most studies were focused on organic mercury (e.g., MeHg) (Liao et al., 2007; Cuello et al., 2012) and/or inorganic HgCl2 (Ismail & Yusof, 2011; Wang et al., 2011; Wang et al., 2013). The toxic potential of α-HgS and β-HgS used in traditional medicines is largely unknown. The present study demonstrated that embryo toxicity followed exposure to mercury, with MeHg the most toxic, followed by HgCl2, while α-HgS and β-HgS had little toxicity.

In humans and rodents, MeHg is known to cross the placenta and reach the fetus where it is responsible for developmental toxicity (Clarkson, Magos & Myers, 2003; Gandhi, Panchal & Dhull, 2013). In laboratory studies using fish, MeHg exposure of early life stages produced developmental toxicity. Exposures of ≤80 ppm (mg L−1) to medaka embryos have increased mortality and caused teratogenic effects including stunted growth, decreased heart rate, and small eyes with reduced pigmentation, among others (Heisinger & Green, 1975; Dial, 1978). In zebrafish (Danio rerio) larvae exposed to ≤25 mg L−1, down-regulation of >70 proteins was associated with morphological changes including but not limited to: smaller swim bladder, unabsorbed yolk, jaw deformities, and bent body axis (Cuello et al., 2012). In the present study, 0.1 µM MeHg produced pericardial edema that in severe cases formed a tube heart, reduced eye pigmentation, and failed swim bladder inflation. Each of these changes could impact the organism’s health and survival (Dial, 1978; Hawryshyn, Mackay & Nilsson, 1982; Marty, Hinton & Cech, 1995).

Compared to MeHg, HgCl2 primarily induces kidney and liver injury in rodents and fish (Klaassen, 2001; Lu et al., 2011a; Wu et al., 2016). However, exposure of mouse- (Van Maele-Fabry, Gofflot & Picard, 1996), sea urchin- (Marc et al., 2002), and medaka embryos (Ismail & Yusof, 2011) to HgCl2 produced developmental toxicity. Wang et al. (2011), Wang et al. (2013) and Wang et al. (2015) studied HgCl2 in adult marine medaka (Oryzias melastigma), and their proteomic analysis showed down-regulation of several dozen proteins including some related to oxidative stress after acute (1,000 µg/L for 8 h) and chronic (10 µg/L for 60 days) exposures. Subsequent work on liver and brain developed a pathway analysis for potential toxicity (Wang et al., 2013; Wang et al., 2015). However, ultrastructural changes consistent with altered cells were more apparent in brain than in liver, where reported alterations of mitochondrial and endoplasmic reticulum were not supported by the figures. Wester & Canton (1992) provided strong evidence for liver toxicity following exposure of adult guppies (Poecilia reticulata) to MeHg (1–10 µg/L for 1 & 3 months). Alterations involved hepatocytes (cell swelling and nuclear pyknosis) and hyperplastic biliary epithelium of the intrahepatic bile duct.

We have shown that medaka have good potential as a model to investigate developmental effects of different forms of mercury. However, in this study, the potential for developmental toxicity of α-HgS and β-HgS proved much lower than that of MeHg or HgCl2. For example, at 10 dpf, less than 5% of medaka embryos died and survivors had no apparent teratogenic effects. These results are comparable with studies of α-HgS and cinnabar-containing traditional medicines in mice (Lu et al., 2011a; Lu et al., 2011b; Wu et al., 2016) and rats (Shi et al., 2011). Similarly, β-HgS has been shown to be much less toxic as compared to HgCl2 in mice (Zhu et al., 2013; Li et al., 2016; Wu et al., 2016). In those studies, α-HgS and β-HgS were administered orally at 1.5–6 fold (mouse studies) and 20 fold (rat study) above clinical doses, still fourfold higher than the Chinese Pharmacopoeia Allowable Limit (Shi et al., 2011). A recent study in mice showed that gestational exposure to low dose α-HgS (10 mg/kg/day, p.o. × 4 weeks) resulted in offspring with severe neurobehavioral dysfunctions (Huang et al., 2012).

The present study used aqueous exposure of embryonated eggs, which brings up bioavailability related to solubility and the role of the chorion. The solubility of cinnabar is known to be quite low (<0.001 g/L at 20 °C), but preparations described by Liu et al. (2008), used in the present study, can increase this value. However, future work will need to describe how formulation of the intended medicinal end product affects the solubility of this mineral (Kamath et al., 2012). The chorion, a semi-permeable membrane, provides a degree of protection from its surrounding environment (Villalobos et al., 2000) and, near time of hatching, becomes more permeable (Hamm & Hinton, 2000). Because xenobiotics in general, and more recently nano-metals, have been shown to enter through chorion pore canals, this route can affect developing embryos (Villalobos et al., 2000; González-Doncel et al., 2003; Wu & Zhou, 2012). Future work is needed to compare if and how different forms of mercury penetrate the chorion.

Mercury compounds display multiple organ toxicity (e.g., hepatotoxicity, nephrotoxicity and neurotoxicity) in adult humans and experimental animals (Klaassen, 2001; Liu, Goyer & Waalkes, 2008; Lu et al., 2011b; Shi et al., 2011). One of the most common mechanisms for toxicity is oxidative stress. For example, mercury induces the production of reactive oxygen species (ROS) by binding to intracellular thiols (GSH and sulfhydryl proteins) and by acting as a catalyst in Fenton-type reactions, producing oxidative damage (Klaassen, 2001; Liu, Goyer & Waalkes, 2008). Heme oxygenase-1 (Ho-1) is an oxidative stress biomarker and was one of the most sensitive genes in response to toxic stimuli in a study of zebrafish embryos acutely exposed to 14 different chemicals (Weil et al., 2009). In the present study, MeHg increased Ho-1 by 6-fold and HgCl2 by 2.3-fold compared to controls, suggesting that MeHg produced more oxidative damage to embryos. The lack of increased Ho-1 expression by α-HgS and β-HgS coincided with the observed low developmental toxicity. This is in agreement with rodent studies that showed exposure to α-HgS (300 mg/kg) did not induce Ho-1 in liver or kidney, concordant with less hepato- (Lu et al., 2011a) and nephrotoxicity (Lu et al., 2011b).

Metallothionein (MT) is thought to protect against oxidative stress and detoxify heavy metals including mercury (Klaassen, Liu & Choudhuri, 1999). Induction of MT by mercury is a sensitive biomarker for exposure in a variety of fish species (Van Cleef-Toedt, Kaplan & Crivello, 2001; Cheung, Lam & Chan, 2004; Chan et al., 2006; Oliveira et al., 2010), including Javanese medaka (Oryzias javanicus) (Woo et al., 2006). We found MT increased by 4–6 fold compared to controls with MeHg and HgCl2 exposure but were unchanged following α-HgS and β-HgS. This is similar to rodent studies (Lu et al., 2011a; Shi et al., 2011). It is possible that the increases we observed were due to the timing of our sampling (3 day post exposure). That said, considerable development occurs over 3 days, including formation of many of the major organs (Iwamatsu, 2004). In zebrafish, MT levels have been shown to have strong and ubiquitous expression during early embryonic development and drop off later, and it is highly susceptible to metals (Chen et al., 2004). The displacement of essential metals by Hg may compromise multiple cellular processes (Amiard et al., 2006), likely problematic during periods of rapid cell division and differentiation. At this point, we cannot directly link the morphological changes we observed to MT, but simply state that the observed increase was a response to the added mercury. The induction of MT by MeHg and HgCl2 reinforces the importance of this biomarker of mercury compounds in general, and identifies it as a potential biomarker for developmental toxicity. Future work with these compounds at intermediate concentrations may provide enough surviving embryos to gauge stage specific gene expression changes in response to various mercury compounds.

Overall, this study confirmed that the medaka embryo assay is a useful tool for determining and comparing potentials of developmental toxicity for various forms of mercury. We found survival even at middle concentrations of the more toxic forms, suggesting that our ranges were acceptable given the design of our study. We did not observe changes with α-HgS and β-HgS, possibly due to the limitations in concentrations resulting from the initial constraint of comparing mercurials at the same concentrations. The rapidity, repeatability, broad salinity tolerance, and precision of this model will enable assessment of a broad range of formulations, concentrations, and mechanisms in the future.

Conclusions

The current study evaluated medaka embryo toxicity caused by exposure to MeHg and HgCl2 and compared results to mercuric sulfides (α-HgS and β-HgS) used in traditional medicines. MeHg and HgCl2 caused increased mortality and developmental toxicity. The latter presented as pericardial edema that, in severe cases, resulted in a tube heart, reduced eye pigmentation, and failure to inflate the swim bladder. Developmental toxicity appeared to be in the order of MeHg > HgCl2 ≫ α-HgS = β-HgS, indicating that the chemical forms of mercury were a major determinant of its toxicity to medaka embryos. While this work only involved two medicinal formulations, these assays will be useful in the study of other permutations of cinnabar-based medicinals and their toxic mechanisms.

Supplemental Information

Table S1 Public health guidelines for mercury in drinking water and fish: creating context and justification for this study

Click here for additional data file.

Table S2 Mean survival and mortality (±SD) for each concentration of the mercury compounds tested

Click here for additional data file.

Additional Information and Declarations

Competing Interests

Author Contributions

Animal Ethics

Data Availability

Jie Liu is an Academic Editor for PeerJ.

Wu Dong and Jie Liu conceived and designed the experiments, performed the experiments, analyzed the data, contributed reagents/materials/analysis tools, wrote the paper, prepared figures and/or tables, reviewed drafts of the paper.

Lixin Wei conceived and designed the experiments, contributed reagents/materials/analysis tools, reviewed drafts of the paper.

Yang Jingfeng analyzed the data, contributed reagents/materials/analysis tools, reviewed drafts of the paper.

Melissa Chernick wrote the paper, prepared figures and/or tables, reviewed drafts of the paper.

David E. Hinton contributed reagents/materials/analysis tools, wrote the paper, prepared figures and/or tables, reviewed drafts of the paper.

The following information was supplied relating to ethical approvals (i.e., approving body and any reference numbers):

Medaka (Oryzias latipes) were maintained at the Duke University Aquatic Research Facility under standard recirculating water conditions following approved animal care and maintenance protocols. Duke University Institutional Animal Care and Use Committee, No. A062-15-02 for breeding and No. A031-15-01 for toxicity testing protocol.

The following information was supplied regarding data availability:

Figshare: https://figshare.com/s/23641deb4c3e10e09259;

https://figshare.com/s/3ef822119d859083275d;

https://figshare.com/s/7ae5a4e6a4cbfec828a6;

https://figshare.com/s/469c738bbda93b93873b;

https://figshare.com/s/60fad618f8beb67696b9;

https://figshare.com/s/ba2608b17e612ca244b7.

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
