# Peer review of "Developmental toxicity from exposure to various forms of mercury compounds in medaka fish (Oryzias latipes) embryos"

_PeerJ, doi:10.7717/peerj.2282_

## Round 0.1 · original submission · Major Revisions

Please include in your response to review a statement of exactly what you changed in the manuscript, for each reviewer point. Also, please note the comments regarding plagiarism. This must be addressed.

Reviewer 1 ·

Basic reporting

English is sometimes quite poor. For instance:

« MeHg proved to be highly toxic, at 1-10 µM concentration, all embryos died before 3 days post-fertilization (dpf), and 0.1 µM produced some embryos died by 7 dpf and 30% mortality occured at 10 dpf. HgCl2 is also toxic to medaka embryos, at 10 µM of HgCl2, all embryos died before 4 dpf, and at 1 µM of HgCl2 40% mortality was seen at 10 dpf, 0.1 µM HgCl2 produced 12% mortality at 10 dpf. »

« No developmental abnormalities at 5 dpf embryos treated with 10 µM a-HgS and b-HgS. »

It is also the case for the (very long) last sentence of the discussion.

The transition between sentences is sometimes not very clear:

« Mercury vapor (Hg0) evaporates from water, soil, volcano eruption/ash. Human activities such as coal combustion, and metal mining industries are globally distributed. Eventually, the released mercury is oxidized to a water-soluble inorganic form,… »

There are sentences in the abstract/introduction that are absolutely not useful to the story:

« Mercury… can become incorporated in marine fish and be a dietary concern for women of reproductive age. » (see later for comment on clarity of the research question)

« Recent studies found that two gene clusters, hgcA and hgcB in microbes, are required for mercury methylation (Parks et al. 2013). »

Fifth paragraph of the discussion (« Mercury-based herbo-metallic preparations… ») should rather be in the introduction section.

A few minor orthographic errors:

e.g. : « severe », « cinnabar »,…

Plagiarism:

« Fish, particularly small sized ones have been the most popular choice for vertebrate test organisms as they are the best-understood organism in the aquatic environment. » (Ismail & Yusof 2011 - first sentence of the introduction)

Literature sometimes not at all relevant:

« Methylmercury is generally more toxic than inorganic mercury to human, aquatic organisms and marine ecosystems (Karimi et al. 2012). » (this article deals with comparisons of Hg concentrations in seafood items)

« Tests using these model fishes are short termed, requiring only a few days to weeks and could be a good substitute for life-cycle tests (Ismail & Yusof 2011). » (this is a reference therein that should be reported)

« Many studies (Liao et al., 2007; Cuello et al., 2012; Gandhi et al., 2013) have been devoted to the toxicity of mercury to various marine animals,… » (two of these studies are involving the use of zebrafish or rat)

There is no mention of a previous study that has investigated the developmental toxicity of mercury in medaka (Dial NA, Teratology, 1978).

As a general comment throughout the manuscript, the authors should realize that (for example) zebrafish is not a marine organism. In the vast majority of cases when « marine » is used, it should be « aquatic » instead.

I believe the structure of the manuscript conforms to PeerJ standard.

Figures are not so relevant.

First, although it is not a firm need, it would have been better perhaps to express data from the first figure with one graph per chemical dose (rather than per mercurial). In this case, DMSO controls could not be shown to improve clarity. This way, a direct comparison of the data between mercurials is possible. Also, I think the manuscript would benefit from deriving ECx values for the parameters measured (see below).

Second, the third figure absolutely needs to be re-done. This is because gene expression levels are expressed as percentages of the Ct value of the reference gene. This is not the way to do it (you cannot obtain ratios of Ct values to get a sense of gene expression levels as Ct are exponential values). Rather, as detailed in the article that the authors are refering to, the methodology is that of delta delta Ct. It follows of course that statistics and conclusions should be revised for the manuscript.

Experimental design

As stated above, it feels that the research question could be made clearer if the authors avoid the use of sentences like:

« Mercury… can be a dietary concern for women of reproductive age. » (first sentence of the abstract)

« Consumption is the major route of exposure to MeHg… »

Rigorous investigation : not so sure, considering the mode of data analysis for qPCR data, and also the fact that cinnabar and metacinnabar have poor solubilities and intestinal absorption capacities (at least in mammalian models). Regarding the latter, the authors should comment on the low solubility/bioavailability of these compounds and the validity of testing them at micromolar doses (i.e., were they soluble at these doses in their experimental medium ?). Also, I would think that deriving EC values would be a good thing with the data obtained, and should also help in the discussion section. Indeed, instead of stating « The present study showed that developmental toxicity potentials of a-HgS and b-HgS are at least 100 fold lower than MeHg, and at least 10 fold lower than HgCl2 », a more precise and informative way of comparing effects of mercurials would be to say « Based on ECx values, cinnabars are x fold less effective at inducing a particular phenotype than MeHg and x fold less than HgCl2 ». Lastly, the authors do not provide a rationale for conducting gene expression studies at 3 dpf based on morphological observations at 10 dpf (although it is clear that severe phenotypes were seen at 5 dpf), and it is also true for the doses used (again, based on EC values, the use of more relevant effective doses could have been used to assess Ho-1 and MT induction).

Regarding the qPCR studies as they are explained in the material and methods section, I cannot be sure that they follow the MIQE guidelines. What about the efficiency of amplification with the primers used? What about the specificity of amplicons (based on sequencing)? What about the stability of Ct values for the reference gene?

There is no approval reference number mentioned for the research project.

Validity of the findings

This study could certainly be made more impactful providing that the authors report EC values (allowing straight comparisons between mercurials) and convince the reader that cinnabar concentrations in media did not reach solubility saturation at any tested dose.

The discussion is not much more than a re-description of the results.

Regarding the statistical validity of the observations: it is not clear to me if the studies were conducted on multiple batches (i.e., biological replication) of developing medaka or 10-20 individuals from a single batch. It would be essential to know this prior to decide on the acceptability of the data (statistical validity).

It would have been benefitial to include more gene in qPCR studies (e.g., HSP70, …), and use the data in determination of EC (induction) values. But this is just an opinion.

Additional comments

The interest of the study is mainly to provide comparative toxicity data for cinnabars and other mercurials. There is not much data provided by this study, except morphological observations of general toxicity at 10 dpf. Also, we need to be convinced that the cinnabar doses applied to the media are made available to developing medaka (i.e., did not reach solubility saturation). The English as well as the science behind the manuscript (including literature survey) should be thoroughly revised.

Reviewer 2 ·

Basic reporting

Basic Reporting
• The submission must adhere to all PeerJ policies (see: 'Journal Policies').

- The raw data were provided in Figshare for the PCR analyses only. Figshare had only the same figures as appeared in the manuscript for survival (Fig 1), and developmental abnormalities (Fig 2).

• The article must be written in English using clear and unambiguous text and must conform to professional standards of courtesy and expression.

- The article is in English, but certain sections are not clearly written and would benefit greatly from revision by a native English speaker. See pdf for specific areas.

• The article should include sufficient introduction and background to demonstrate how the work fits into the broader field of knowledge. Relevant prior literature should be appropriately referenced.

- The article appears to have been hastily written and many critical details are missing. For example, the authors do not provide any context or justification for the doses tested, why the Stage 10 life stage was chosen to initiate the experiment, why the experiment was carried out for 10 days, how many animals were exposed per well, what the data points represent in Figure 1 (are these means? Medians? Why are there no error bars or confidence intervals?), 15-18 embryos were evaluated for developmental effects but no quantitative data are presented on the percent with each type of abnormality and no statistical analyses are presented to support their statement that treatment differences were significant, what are the relative environmental concentrations of the 4 forms of mercury they tested (i.e. the relative occurrence, and thus significance, in the environment).

• The structure of the submitted article should conform to an acceptable format of ‘standard sections’ (see our Instructions for Authors for our suggested format). Significant departures in structure should be made only if they significantly improve clarity or conform to a discipline-specific custom.

-Structure adhered to typical journal standards for manuscripts of this type.

• Figures should be relevant to the content of the article, of sufficient resolution, and appropriately described and labeled.

-The 3 figures are relevant, of sufficient resolution, and appropriately labeled however they are not appropriately described in the figure legends or the text. The text and legends lack sufficient detail to permit this reviewer to fully evaluate the author’s interpretation of the data

• The submission should be ‘self-contained,’ should represent an appropriate ‘unit of publication’, and should include all results relevant to the hypothesis. Coherent bodies of work should not be inappropriately subdivided merely to increase publication count.

- The work presented is likely sufficient for publication, but due to the lack of detail in the Methods (e.g. power analysis of variability to determine the number of replicates needed to identify and Results (e.g. power analysis, means with error bars, it is difficult to determine if the replication is sufficient for statistical rigor.

Experimental design

• The submission must describe original primary research within the Aims & Scope of the Journal.

- Article meets these parameters

• The submission should clearly define the research question, which must be relevant and meaningful. The knowledge gap being investigated should be identified, and statements should be made as to how the study contributes to filling that gap.

- The authors describe the knowledge gap, and how this study addresses part of that gap, but not until the Discussion. This point should be made in the Introduction to justify the study.

The authors also need to provide context, and justification, for the concentrations chosen for the study. How do these compare to drinking water guidelines for methylmercury?

• The investigation must have been conducted rigorously and to a high technical standard.

- It is impossible to judge the rigor of the study because the authors do not provide sufficient detail in many areas. See the BASIC REPORTING section:

-…..the authors do not provide any context or justification for the doses tested, why the Stage 10 life stage was chosen to initiate the experiment…..[see BASIC REPORTING section for remaining comments]

• Methods should be described with sufficient information to be reproducible by another investigator.

Methods are not sufficiently described.

1) It is not clear how many embryos were used. In the Methods, the authors state 15-18 embryos were used for the mortality and teratogenesis studies, but in the legend of Figure 1, 18-20 embryos are listed. So the actual number of embyros used for mortality and teratogenesis is unclear. It is also not stated if they used 1 embryo per well in the 6 well plates, or multiple embyros per well. At 1 embryos per well: [18 embryos/treatment ] / 6 wells per plate = 3 plates per treatment x 6 doses/treatment x 4 mercury treatments = 72 plates would have been used.

Also, it appears that the treatment was static, as the authors do not indicate renewal of the test solutions. Other researchers testing methylmercury toxicity with fish embryos renew test water in their multiwall plates every 24 h (e.g. Ismail & Yosof 2011 citation in the present study); which is likely due to metal concentrations not being stable in solution over time, especially in dilute solutions. If solutions were not renewed on a near daily basis, it is likely that concentrations fell over the first few days of the experiment, and may not have fallen at the same rate for each of the 4 mercury forms. This complicates interpretation of the relative toxicity of the 4 mercury forms. For the mRNA data, which are based on a 3 day exposure, it is likely that the changes in HO-1 and MT reflect initial response of these genes to exposure to these chemicals. These issues should be discussed in the Discussion in terms of how they affect interpretation of the data.

2) How were the homogenates for PCR prepared? Were all embryos in a single treatment pooled for a single homogenate, which was then split into 3 replicates for PCR analysis? If so, then the authors should use SD, not SEM. SEM is used to describe the precision of the estimate of the population mean (e.g. comparing the variability of 3 pools of 6 embryos per pool); SD is used to describe the variability of the test itself (variability of the PCR test determined by running 3 replicates of 1 pool).

3) Did the authors determine, prior to the experiment, that b-actin does not vary with the mercury compounds used, for this species at this life stage over this 10 day duration? The raw data suggest there is some variation of b-actin, is this variation acceptable? Outlier analysis (Grubbs Outlier test) by this reviewer of the 5 b-actin data points provided in the Figshare raw data file indicates no outliers, which making it likely the variation in b-actin observed among the 5 treatments is not significant. Such an analysis should be done by the researchers and presented to justify the use of b-actin for normalizing the HO-1 and MT mRNA data.


• The research must have been conducted in conformity with the prevailing ethical standards in the field.

- The animals were cultured in accordance with Animal Care and Use protocols approved by the IACUC committee at Duke University.
- However the authors do not explicitly state that the experimental design was approved by this committee. They also do not state how the embryos were euthanized, instead they simply state that the experiments were terminated.
- It is important to note that the INTRODUCTION contains a substantive number of sentences that are taken nearly verbatim from Ismail & Yosof 2011, one of the references cited in the bibliography. This is plagarism.

Validity of the findings

Validity of the Findings
• The data should be robust, statistically sound, and controlled.

Difficult to fully judge the quality of the data because critical information is missing.

Mortality data: Figure 1. Why did the authors choose 15-18 (or 18-20) embryos per exposure for the mortality data? Is this based on a power analysis of previous data on the variability of toxicant-induced mortality for medaka of this life stage? Others have used far more (120 embryos per treatment, Ismail & Yosof 2011, cited by the authors). In Figure 1, the legend states ‘18-20 embryos/exposure’, but there is only 1 data point for each exposure each timepoint, indicating that the data points must be means. If they are means, why are there no error bars? Without error bars, and without evidence of statistical tests (p values), it is not possible to determine if there are statistical differences between doses for any of the 4 mercury compounds tested.

Percentage data must be converted before they can be run through statistical tests, such as ANOVA. [see http://archive.bio.ed.ac.uk/jdeacon/statistics/tress4.html] for a straightforward explanation.

A common transformation for percent survival data is arcsine-square root:

Arcsine [sqrt {percent survival/100}] x 2


Developmental toxicity data: Figure 2. It is very clear from the photographs that certain mercury compounds had significant deleterious effects on development. However, the authors do not provide any quantitative information for how many individuals were affected at each concentration. Because the authors did not provide raw data in Figshare for developmental effects, I was unable to determine how strong the data are. At the very least, a table should be provided indicating the number of organisms displaying each abnormality out of the 15-18 embryos tested per treatment.

mRNA data: Figure 3. There were raw data provided for the PCR assay in Figshare, the data are provided as means with error bars, and statistical outcomes are provided. However, the authors did not indicate that they had determined prior to their study (either from the literature or in their laboratory) that b-actin is not altered by exposure to mercury at these concentrations, in these life stages, over this time frame. As has been known for many years that housekeeping genes, including b-actin, are not always stable in the face of perturbations and the current practice is evaluate the response of the housekeeping gene prior to use. Outlier analysis (Grubbs Outlier test) by this reviewer of the 5 b-actin data points provided indicates no outliers, which indicates it is likely the variation in b-actin observed among the 5 treatments is not significant. The authors need to provide such information to justify their use of b-actin.

I refer the authors to this publication:

Arukwe A. 2006. Toxicological housekeeping genes: Do they really keep the house? Environ Sci Technol 40:7944-7949.

• The data on which the conclusions are based must be provided or made available in an acceptable discipline-specific repository.

The raw data are only provided in Figshare for the mRNA PCR assay; raw data files are missing for mortality and for developmental toxicity. Instead, the figures from the manuscript are simply repeated in Figshare, no additional information is provided.

• The conclusions should be appropriately stated, should be connected to the original question investigated, and should be limited to those supported by the results.

The conclusions are appropriate, except for two statements in the Discussion:

The present study showed that the developmental toxicity potentials of α-HgS and β-HgS are at least 100-fold lower than MeHg and at least 10-fold lower than HgCl2.
(and a similar statement that appears elsehwhere in the Discussion)

The authors do not explain what these fold differences are based on. There are no quantitative data, or even a relative scale of severity of effects, presented for the teratogenesis data. If the authors are basing these fold differences on no effects at 10 uM α-HgS and β-HgS compared to ANY SIGN of teratogenesis at 0.01 uM MeHg and 0.1 HgCl2, this needs to be clearly stated.

Thus, similar to our prior…..induction of MT …..could be an important biomarker for developmental toxicity of mercury compounds.

The present study was not designed to determine if the MT gene regulates development. The finding that MT is elevated by mercury only indicates that it is a good biomarker for exposure to certain metals that are know to regulate MT (e.g. Cd, Hg, Zn). A study designed to determine if MT is a good biomarker for developmental effects would need to determine if development was altered in organisms where the MT gene was suppressed or knocked out. This statement must be modified to indicate the MT gene in early life stage medaka is sensitive to mercury, and then compare that to whether others found MT to be sensitive to Hg or other metals in early life stage fish.

• Speculation is welcomed, but should be identified as such.

There were no speculative statements in this article.

• Decisions are not made based on any subjective determination of impact, degree of advance, novelty, being of interest to only a niche audience, etc.

No such decisions are in the present article

• Replication experiments are encouraged (provided the rationale for the replication, and how it adds value to the literature, is clearly described); however, we do not allow the ‘pointless’ repetition of well-known, widely accepted results.

This reviewer found no evidence that these studies duplicate those of others


• Negative / inconclusive results are acceptable.

The authors have mRNA data for HSP in the raw data file, but they do not present it. It is unclear why they are not reporting the HSP results… there is no clear response of this gene to mercury, which may make it a useful housekeeping gene…..it should be published.

Additional comments

This paper has some interesting findings, but critical information is missing from the Methods and from the Results and additional statistical analyses will greatly strengthen the findings. The Discussion covers a broad arena, but it is important to give the concentrations, life stages and durations of the experiments you are comparing your data to, as well as the endpoints, otherwise the data cannot be fully interpreted (they may have seen effects like yours, but in different life stages or at concentrations far higher than yours. You may miss impt conclusions this way. Avoid plagarism (Introduction has several such sentences). I urge you to include the HSP data.

Annotated reviews are not available for download in order to protect the identity of reviewers who chose to remain anonymous.

Reviewer 3 ·

Basic reporting

The present study demonstrates the developmental toxicity of MeHg and three forms of inorganic mercury in medaka embryos. The focus of most other studies is largely on methyl mercury; relatively few address inorganic Hg toxicity. This study attempts to fill that gap.

Experimental design

Do inorganic and organic forms of mercury cross the chorion at the same rate? What is the evidence that all the compounds tested here actually were able to get through this barrier? No data are presented that confirm that these mercurials actually were present in the embryos.

Validity of the findings

Results
Pg. 12 “MeHg (4-fold) and HgCl2 (5-fold) increased MT mRNA expression, while α-HgS (1.4-fold) and β-HgS (1.3-fold) had no appreciable effects. Ho-1 is a widely used biomarker for oxidative stress, and MeHg and HgCl2 (2.5-fold) increased Ho-1 mRNA expression by 6-fold and 2.3-fold, respectively, while α-HgS (1.8-fold) slightly increased Ho-1 expression, while β-HgS (1.4-fold) had no significant effects.” These statements are not clear. For example, it reads that a 4-fold concentration of MeHg is used – or does MeHg increased expression of MT mRNA 4-fold over control? At what concentration of MeHg?

Discussion

Neither medaka nor zebrafish are marine animals.

“Thus, MeHg is teratogenic not only to humans and mammals, but also to teleost embryos of marine or fresh water habitats.” No evidence is presented here that this is a universal phenomenon in all freshwater and marine fish.

Fig. 1 “Survival” is misspelled in all panels. Remove the symbol key from each panel and put it into the figure legend. Label panels A,B,C, etc., and identify the compound used in the legend.
It is difficult to impossible to distinguish all the symbols in these figures as they overlap in many cases.
“Mortality of mercury compounds in medaka fish embryos” – implies that mercury, not the fish, is dying; reword.
Replace “Data are…” with “N =”

Fig 2 replace “madaka” with “medaka” (and check throughout the text); “sever” should read “severe”.
Magnification bar is not present in every panel.
Remove extraneous text from the panels and put it into the figure legend.
These are representative picture of how many replicates?

Additional comments

An interesting paper but with numerous misspellings, grammatical errors and awkward sentence structure which at times leads to misinterpretation of the data. Manuscript needs careful and extensive editing.

There are a number of points that should be addressed that are discussed above..

---

## Round 0.2 · Major Revisions

Dear Dr. Dong,

The revised manuscript has been reviewed again by one of the original reviewers. There are serious concerns that have not been met. Your response to review must include a careful reading of the current review, and the prior reviews, by ALL authors. All authors must agree with the second revision to manuscript.

Please include a point by point response of the changes made to the manuscript for each point of the reviewer.

I again draw your attention to the following note: "PeerJ does not offer copyediting, so please ensure that your revision is free from errors and that the English language meets our standards: uses clear and unambiguous text, is grammatically correct, and conforms to professional standards of courtesy and expression."

I urge you to have all authors read the manuscript carefully one more time to ensure that the English is as good as it can be.

Sincerely - John Stegeman

Reviewer 2 ·

Basic reporting

See the attached PDF

Experimental design

See the attached PDF

Validity of the findings

See the attached PDF

Additional comments

See the attached PDF

Annotated reviews are not available for download in order to protect the identity of reviewers who chose to remain anonymous.

---

## Round 0.3 · Minor Revisions

Dear Dr Dong,

Please be sure to have Dr. Hinton see these reviewer comments and participate in the revision.

Reviewer 2 ·

Basic reporting

See general comments to the authors

Experimental design

See general comments to the authors

Validity of the findings

See general comments to the authors

Additional comments

REVIEW of version 2 [5-19-2016] of Dong et al. for Peer J

The manuscript is much improved. There are just a few lingering issues that should be addressed, after which I consider it acceptable for publication.

I do not need to see the final version.

Methods

Line 151. Further explanation of concentration choice is needed. For instance:

Concentrations were chosen to provide exposure levels that produced developmental abnormalities without leading to 100% mortality and were based on preliminary range finding assays in the same design.

Line 208. ‘The mortalities produced by embryo exposure to 1 and 10 μM…’.

Line 263. To facilitate comparison of exposure concentrations in your experiment to concentrations used by others, it is imperative that you put them in common units.
For 80 ppm, put the mg/L equivalent in parenthesis:
‘…….≤80 ppm (XX mg L-1)…’

Line 266. Be clear. Is (>70) the number of proteins affected? If so, write this explicitly:
‘ Down-regulation of proteins (> 70 proteins) and resultant…’


Figure 1. Legend.
The authors should not be pooling all the data –(n=30 to 54); they lose the point of doing replicates. Perhaps they mean a TOTAL of 30-54 embryos/group? But what they should put in the legend is this: Each data point represents the MEAN of n=3 replicates of 10-18 embryos per replicate.

Figure 2. Legend.
Legend still contains RESULTS “Reduced eye pigmentation, as well as delayed or arrested growth were observed in 5 dpf embryos with MeHg and HgCl2 (B-C). Pericardial edema was evident in 0.1 µM MeHg exposed individuals (B, E), the severity resulting in a tube heart (arrows). Absence of an inflated swim bladder was observed in MeHg and HgCl2 exposed fish (E-F) but not α-HgS or β-HgS (G-H).” This should be removed from the legend and put into the RESULTS section.

Figure 3. Legend.
‘Gene expression analysis of mercury compounds in medaka embryos’
You are not analyzing the gene expression of mercury compounds. You are analyzing the expression of genes in embryos exposed to mercury compounds….

Reviewer 3 ·

Basic reporting

The authors have extensively revised the original manuscript, carefully responding to the reviewers’ comments. The paper is now more concise, grammatically correct and the methods now contain details which were previously missing. Minor errors & some suggestions are noted below. Note that my "red" text did not make it into this pasted document, so I added brackets [ ].


Abstract
37 ….toxicity, [m]etallothionein, heme oxygenase-1, mercury….

Intro
103 cysteine-rich, metal binding proteins that detoxify excess heavy metal ions and [play] a general role in….

Methods
180 5#- use prime “ ‘ ”not “#” i.e., 5’,3’
197 actin data [were] analyzed by a Grubbs Outlier test; outliers did not alter the results ….

Results
Present means and standard deviations in a supplementary table.

Fig 1 A. It appears from the graph that fish exposed to 0.0001ug MeHg had ~ 100% survivability by dpf 10 – higher than the control animals.
No need to repeat percentage (%) on Y axis label or DPF on x axis.
Move symbol key into the figure legend, not on the graph itself (1D).

Fig 2 Red arrows are difficult to see against background; may not be visible to anyone with red-green color blindness. Symbols “e” and “S” are italicized in legend but not on figure. Could omit the dpf on figures as it is clear in the legend.

Fig. 3 Use same sample shading patterns for same mercurial compounds in both bar graphs.

Discussion
274 Compared to MeHg, HgCl2 primarily induces kidney and liver injury (Klaassen, 2001; Lu et al., 2011a; Wu et al., 2016) - In what species?

312 needed [to] compare if and how different forms of mercury penetrate the chorion

Throughout – order references in chronological order when citing more than one

Experimental design

This section was much improved.

148 There was no control that did not contain DMSO – what is the evidence that DMSO does not contribute to the toxicity? Does each treatment contain the same concentration of DMSO?

Concentrations were chosen based on preliminary range finding assays in the same design. But what is the biological reason for doing so?
The fact that the test solutions were not renewed during the experimental period is a bit concerning – it is likely that the embryos/fish are not seeing the described dose during the entire treatment period.

Validity of the findings

Discussion has been much improved.

---

## Round 0.4 · accepted · Accept

Dear Drs. Dong and Hinton,

Please note that I am sending a version with some text edits that you should include when you correct the proof copy from the publisher.

Thanks for the effort on this manuscript!

John Stegeman